# Correlation between higher-order aberration and photophobia after cataract surgery

**Naoko Ishiguro**[1], **Hiroshi Horiguchi**[1]*, **Satoshi Katagiri**[1], **Takuya Shiba**[2], **Tadashi Nakano**[1]

**1** Department of Ophthalmology, The Jikei University School of Medicine, Tokyo, Japan, **2** Roppongi Shiba Eye Clinic, Tokyo, Japan

\* hhiro@jikei.ac.jp

**Data Availability Statement:** Data cannot be shared publicly because of ethical restrictions. Data are available from the Institutional Review Board of the Jikei University School of Medicine for researchers who meet the criteria for access to

## Abstract

Cataract surgery impinges on the spatial properties and wavelength distribution of retinal images, which changes the degree of light-induced visual discomfort/photophobia. However, no study has analyzed the alteration in photophobia before and after cataract surgery or the association between retinal spatial property and photophobia. Here, we measured the higher-order aberrations (HOAs) of the entire eye and the subjective photophobia score. This study investigated 71 eyes in 71 patients who received conventional cataract surgery. Scaling of photophobia was based on the following grading system: when the patient is outdoor on a sunny day, score of 0 and 10 points were assigned to the absence of photophobia and the presence of severe photophobia prevents eye-opening, respectively. We decomposed wavefront errors using Zernike polynomials for a 3-mm pupil diameter and analyzed the association between photophobia scores and HOAs with Spearman's rank sum correlation (rs). We classified patients into two groups: photophobia (PP) unconcerned included patients who selected 0 both preoperatively or postoperatively and PP concerned included the remaining patients. After cataract surgery, photophobia scores increased, remained unchanged (stable), and decreased in 3, 41, and 27 cases, respectively. In the stable group, 35 of 41 cases belonged to PP unconcerned. In PP concerned, there were significant correlations between photophobia score and postoperative root-mean-square values of total HOAs (rs = 0.52, p = 0.002), total coma (rs = 0.52, p = 0.002), total trefoil (rs = 0.47, p = 0.006), and third-order group (rs = 0.53, p = 0.002). In contrast, there was no significant correlation between photophobia scores and preoperative HOAs. Our results suggest that the spatial properties of retinal image modified by HOAs may affect the degree of photophobia. Scattering light due to cataracts could contribute to photophobia more than HOAs, which may mask the effect of HOAs for photophobia preoperatively.

## Introduction

Cataract is the leading cause of visual impairment worldwide characterized by opacity of the crystalline lens [1]. Aging has been reported as a major risk factor in developed countries [2], along with other factors such as sunlight exposure [3], corticosteroid use [4], trauma [5], and

confidential data. The Jikei University Hospital Ethics Committee Secretariat 3-25-8 Nishi-Shimbashi, Minato-ku Tokyo, Japan, 105-8461 TEL:+81-3-3433-1111.

**Funding:** Our study was supported by the grant of Japan Society for the Promotion of Science (JSPS) KAKENHI (JP18K16939 and 21K09729 to H.H.) and the Charitable Trust Fund for Ophthalmic Research in Commemoration of Santen Pharmaceutical's Founder (H.H.).

**Competing interests:** The authors have declared that no competing interests exist.

other co-existing diseases (e.g. diabetes mellitus and retinal dystrophies [6]). Cataract patterns can be classified into anterior subcapsular, nuclear, cortical, and posterior subcapsular based on the opacification of the lens [7, 8]. This lens imperfection leads to variable alteration of optical properties, which affects spatial property [9] and the distribution of wavelength [10] on the retinal image. Consequently, cataract causes various visual disturbances, including decreased visual acuity and contrast sensitivity, monocular diplopia [11] and triplopia [12], progression of myopia [13], and glare [14].

Glare is a well-known visual disturbance associated with cataract that results from forward scattering of light [14]. It is, however, a more complicated visual disturbance than other cataract-induced visual deterioration because the concept of glare has not been well defined in a generally accepted way, and is an intermix of disability and discomfort glare [15]. Disability glare is generally defined as the loss of retinal image contrast due to intraocular light scatter or straylight [16]. Noninvasive measurement of forward scattering is impossible in vivo, but can be estimated using psychophysical methods [9, 14]. However, discomfort glare, defined as visual discomfort in the presence of bright light sources, is not well understood [17]. Measurements of discomfort glare are currently only self-reported and large inter-individual variations exist, although visual inputs are physically identical [18, 19]. A psychophysical study suggests discomfort glare is closely associated with spatial properties such as simulated glare sources [19], indicating that cataract could cause not only disability glare but also discomfort glare. Similar to discomfort glare, another clinical term, photophobia, associated with light-induced discomfort, has a controversial definition [20–22]. For this study, we accepted Digre's definition of photophobia as a sensory state in which light causes discomfort to the eye or head [22], since this definition almost equates photophobia with that of discomfort glare, or at least includes discomfort glare.

Modern cataract surgery conventionally involves extraction of cloudy lenses and safe insertion of intraocular lenses (IOLs), eliminating the alteration of optical properties due to cataract. Thus, cataract surgery has impinged on the spatial properties and wavelength distribution of retinal images. Reducing the exposure of the retinal images to pattern glare sources could reduce discomfort glare [19]; meanwhile, the increment of short-wavelength light to the retina could evoke photophobia [20]. Furthermore, postoperative inflammation could evoke photophobia [23–25]. However, to date, no report has yet analyzed the alteration of photophobia/discomfort glare/photoaversion before and after cataract surgery or the association between retinal spatial property and photophobia. Spatial properties of retinal images depend on the point spread function (PSF) of ophthalmic optics. Two measurable components of PSF are straylight and PSF core [9]. Stray light is psychophysically estimated and already reported as a source of both disability [14] and discomfort glare [19]. PSF core is physically measured as wavefront aberrations with a double-pass technique using a wavefront aberrometer [9, 26]. Residuals of wavefront aberrations excluding defocus and astigmatism, that are correctable with eye glasses, are higher-order aberrations (HOAs) [27]. Several clinical studies reported the importance of HOAs for visual functions [28–31]. Here, we investigated the relationship between the subjective degree of photophobia and HOAs before and after cataract surgery. The purpose of this study was to clarify changes in these two parameters before and after cataract surgery, along with the associations between them.

## Materials and methods

### Ethical approval

This study was approved by the Institutional Review Board of the Jikei University School of Medicine and was conducted in accordance with the principles of the Declaration of Helsinki

[approval number: 31-041(9540)]. According to the Ethical Guidelines for Medical and Health Research Involving Human Subjects (Japanese Ministry of Health, Labour and Welfare), the requirement for informed consent from each research subject was waived as this study did not involve interventions, utilize human biological specimens, or collect special care-required personal information. Instead, we posted the documents approved by the Institutional Review Board of the Jikei University School of Medicine on the website and details of this research on the bulletin board in our hospital. We guaranteed the participants the right to refuse participation in this study at any point of the study.

## Patient selection and data acquisition

We studied 71 cases who underwent conventional cataract surgery—which involves phacoemulsification and aspiration, and IOL insertion—in the Jikei University Daisan Hospital from December 2014 to December 2015 by a single surgeon (H. H.). Inclusion criteria for this study included patients with adequate quality pre- and postoperative data including age, decimal best corrected visual acuity (BCVA), pupil size, subjective score of photophobias, wavefront aberrations, and use of same one-piece acrylic IOLs (Tecnis® Optiblue®, Johnson & Johnson Surgical Vision, Inc., Santa Anna, CA, USA). Exclusion criteria included patients with any complications until 1 month after surgery, with other co-existing ophthalmic or systemic diseases that affect vision and/or photophobia such as migraine, and who refused to participate. Postoperative data were obtained during 1 month after surgery. Decimal BCVA was measured using the Landolt C chart and converted to logMAR BCVA. Although no standardized method has been established to measure the degree of photophobia quantitatively, we scored photophobia from 0 (patient does not experience photophobia at all) to 10 (patient experiences severe photophobia to open eyes) with reference to the pain scaling score, and interviewed patients before and after cataract surgery as in a previous study [32–34]. With a subjective photophobia scoring sheet (S1 Text), one ophthalmologist asked a patient in a bright exam room, "Please rate the degree of discomfort with the bright light that you experienced under the sun on a scale of zero to ten. A score of zero means that you did not feel discomfort at all, while a score of ten indicates that you felt severe discomfort too much to open your eyes. A score of five means that you felt discomfort moderately".

We measured the wavefront aberration using OPD-scan3® (NIDEK CO., LTD, Tokyo, Japan), decomposed the data using Zernike polynomials for a 3-mm pupil diameter, and obtained output as Zernike coefficients (μm). We analyzed for a 3-mm dilated pupil because we asked the patients to score the degrees of photophobia on a sunny day when the pupils were expected to be constricted.

## Data analysis

We classified the eyes into three groups based on differences from pre- to postoperative subjective photophobia scores: decrement (score reduced by 2 or more), increment (score increased by 2 or more), and stable (others) groups. We calculated the mean of ages and change in logMAR BCVA from before to after surgery in each group and analyzed the differences between any two groups using the Wilcoxon rank-sum test.

Next, we analyzed the relationships between the subjective photophobia scores and the components of HOAs. To clarify the relationships between the photophobia score and HOAs, we classified cases into two groups: PP unconcerned and PP concerned. PP unconcerned included patients who did not experience photophobia in sunny outdoors, both before and after surgery. PP concerned included the remaining patients. Thus, we separately performed correlation analysis for the two groups—all cases and PP concerned cases. We calculated

Spearman's rank correlations between the subjective photophobia score and signed Zernike coefficient for each Zernike term. We also calculated them with absolute Zernike coefficients. In addition, we employed the root mean square (RMS) of wavefront error as follows: total HOAs combining the 3rd to 6th Zernike orders (THOAs: total HOAs), total coma aberrations combining $Z_3^{-1}$, $Z_3^1$, $Z_5^{-1}$, and $Z_5^1$ (total coma), total trefoil combining $Z_3^{-3}$, $Z_3^3$, $Z_5^{-3}$, and $Z_5^{-3}$(total trefoil), total spherical aberrations (TSAs) combining $Z_4^0$ and $Z_6^0$, third order (3rd) covering $Z_3^{-3}$ to $Z_3^3$, forth order (4th) covering $Z_4^{-4}$ to $Z_4^4$, fifth order (5th) covering $Z_5^{-5}$ to $Z_5^5$, and sixth order (6th) covering $Z_6^{-6}$ to $Z_6^6$. Then, we calculated Spearman's rank correlation between the subjective photophobia scores and each RMS. Lastly, we calculated Spearman's rank correlation between subjective photophobia score and the refractive error of sphere or cylinder, pupil size, and age. A p-value of <0.01 was considered statistically significant in correlation analysis.

Furthermore, to address the possibility of misunderstanding our data with a single linear regression analysis alone, a linear mixed model was used for a total of 2 x 3, which is six sets of data; two sets of data for all patients and PP concerned patients, and three sets of data for preoperative, postoperative and both. In the model, subjective photophobia score was the response, RMS of each HOA group (total coma (μm) and total trefoil), gender and age were fixed effects and individuals were the random effect because light sensitivity is associated both with gender [35] and age [36]. A p-value of <0.05 was considered statistically significant in multivariate analysis. The Image Systems Engineering Toolbox for Biology (ISETBIO [37–40]) was used to simulate the PSFs of representative patients. These analyses were performed using MATLAB 9.1.0 (MathWorks Inc., Natick, MA, USA).

## Results

### Patient demographics

In total, we studied 71 eyes in 71 patients who underwent cataract surgery with adequate pre- and postoperative data. When both eyes were examined in the same patient, we used the side of the eyes with larger postoperative RMS because we speculated that the larger RMS could have stronger association with photophobia and the reliability of postoperative measurement is higher than that of preoperative measurements. Thus, we studied 71 eyes of 71 independent patients. The examined (operated) age ranged from 42 to 89 years (mean ± SD; 73.4 ± 10.0 years). Female and male cases accounted for 44 (62.0%) and 27 (38.0%) respectively. The pre- and postoperative logMAR BCVA was 0.20 ± 0.24 (range, −0.08–1.40) and −0.06 ± 0.11 (range, −0.30–0.40), respectively. LogMAR BCVA significantly improved after surgery (P < 0.001). The pre- and postoperative subjective refractive errors were −1.49 ± 3.99 diopters (range, −13.13–4.25) and −0.82 ± 1.31 (range, −5.75–1.63) respectively. The range of subjective refractive errors narrowed postoperatively because of the correction by IOL, although the change was not statistically significant (p = 0.206). These demographic data are summarized in Table 1.

### Pre- and postoperative photophobia scores

The mean pre- and postoperative subjective photophobia scores were 3.2 ± 3.5 and 1.7 ± 2.3, respectively. Postoperative photophobia scores increased in 3 patients compared with preoperative photophobia scores, remained not changed (stable) in 41 patients, and decreased in 27 patients (Fig 1). In the stable group, 35 patients scored 0 for photophobia (patient did not feel any photophobia) both pre- and postoperatively. Alternatively, approximately half of the patients (35/71, 49.3%) were unaffected by cataract-induced photophobia.

**Table 1. Demographic data of this study.**

| Demographic data | Pre-operation | Post-operation | p |
|---|---|---|---|
| Number of subjects | 71 | | |
| Age (years) | 73.4 ± 10.00 | | |
| Range | 42–89 | | |
| Sex (female/male) | 44/27 | | |
| LogMAR BCVA | 0.20 ± 0.24 | −0.06 ± 0.11 | <0.001 |
| Range | −0.08–1.40 | −0.30–0.40 | |
| Manifest refractive error | −1.49 ± 3.99 | −0.82 ± 1.31 | 0.206 |
| Range | −13.13–4.25 | −5.75–1.63 | |
| Photophobia score | 3.2 ± 3.5 | 1.7 ± 2.3 | 0.077 |
| Range | 0–9 | 0–8 | |

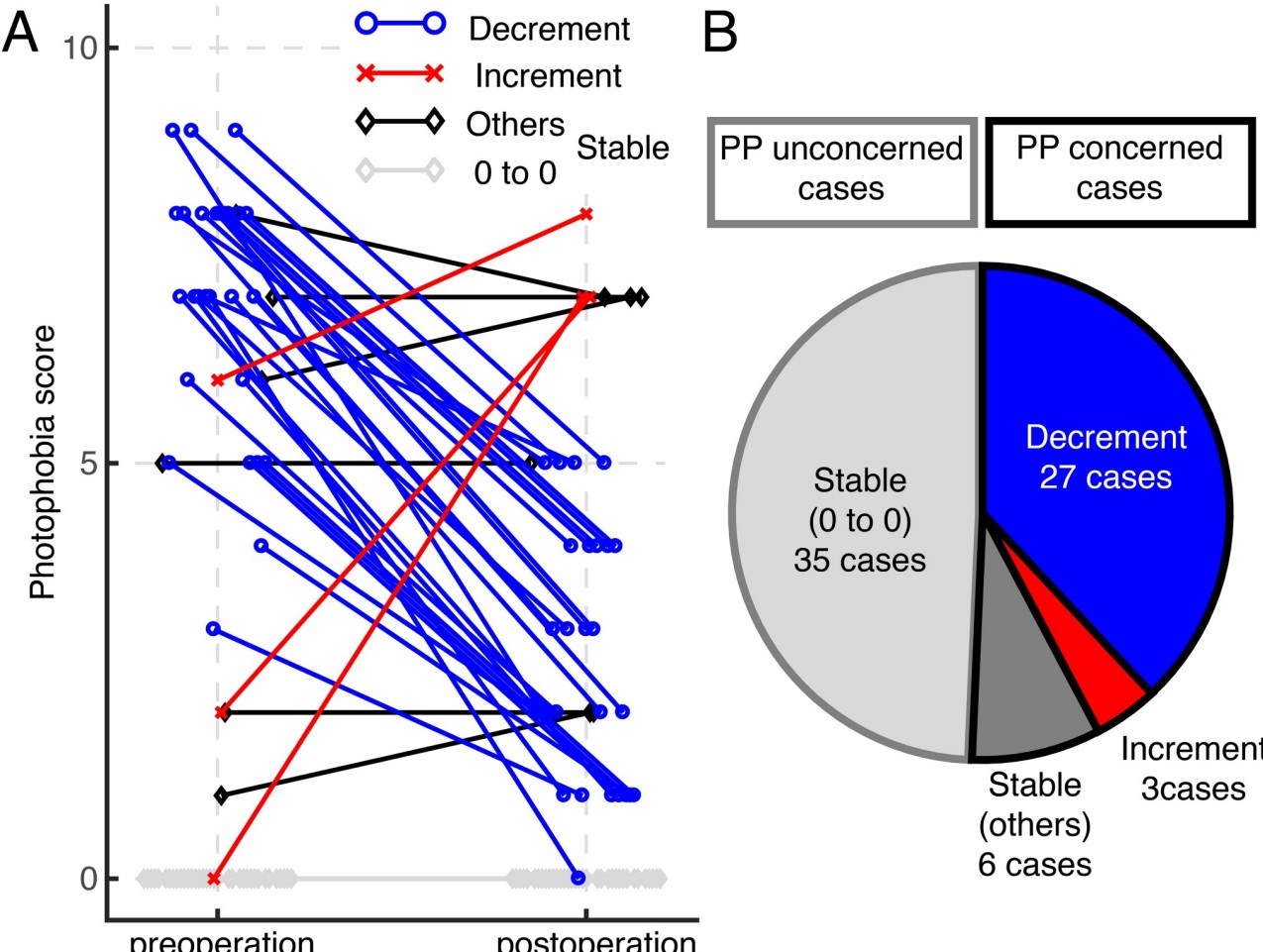

**Fig 1. Preoperative and postoperative subjective photophobia scores.** Fig 1A shows a line graph indicating raw data of pre- and postoperative photophobia scores in 71 patients. The vertical axis represents photophobia scores, ranging from 0 (patient does not experience photophobia at all) to 10 (patient experiences severe photophobia during eye opening). Decrement cases are shown in blue lines with circles, stable cases in black lines with diamonds, and increment cases in red lines with crosses. Fig 1B shows a pie graph indicating the distribution of each group. Twenty-seven patients belonged to the decrement group, 41 to the stable group, and 3 to the increment group. In the stable group, 35 patients showed a photophobia score of 0 both preoperatively and postoperatively, and were considered PP unconcerned cases (light gray outlined). Meanwhile, the remaining cases were considered PP concerned cases (black out lined).

## Preoperative and postoperative wavefront errors

Wavefront errors for a 3-mm-diameter pupil of 62 eyes in 62 patients were analyzed after excluding 9 eyes with inaccurate OPD scan measurements (fitting error > 0.5 μm with 6th Zernike polynomials). Fig 2 shows signed and absolute Zernike coefficients of each Zernike term in 62 eyes with pre- and postoperative measurements. Distributions of postoperative coefficients were generally smaller than those of preoperative coefficients (Fig 2A). With increasing order of the Zernike mode, the coefficients were gradually decreasing in both pre- and postoperative measurements. More than half of absolute Zernike coefficients preoperatively were larger than those postoperatively (Fig 2B).

## Correlation between photophobia scores and wavefront errors

We analyzed the association between subjective photophobia scores and Zernike coefficients. After excluding PP unconcerned patients, which include 3 eyes with inaccurate OPD scan

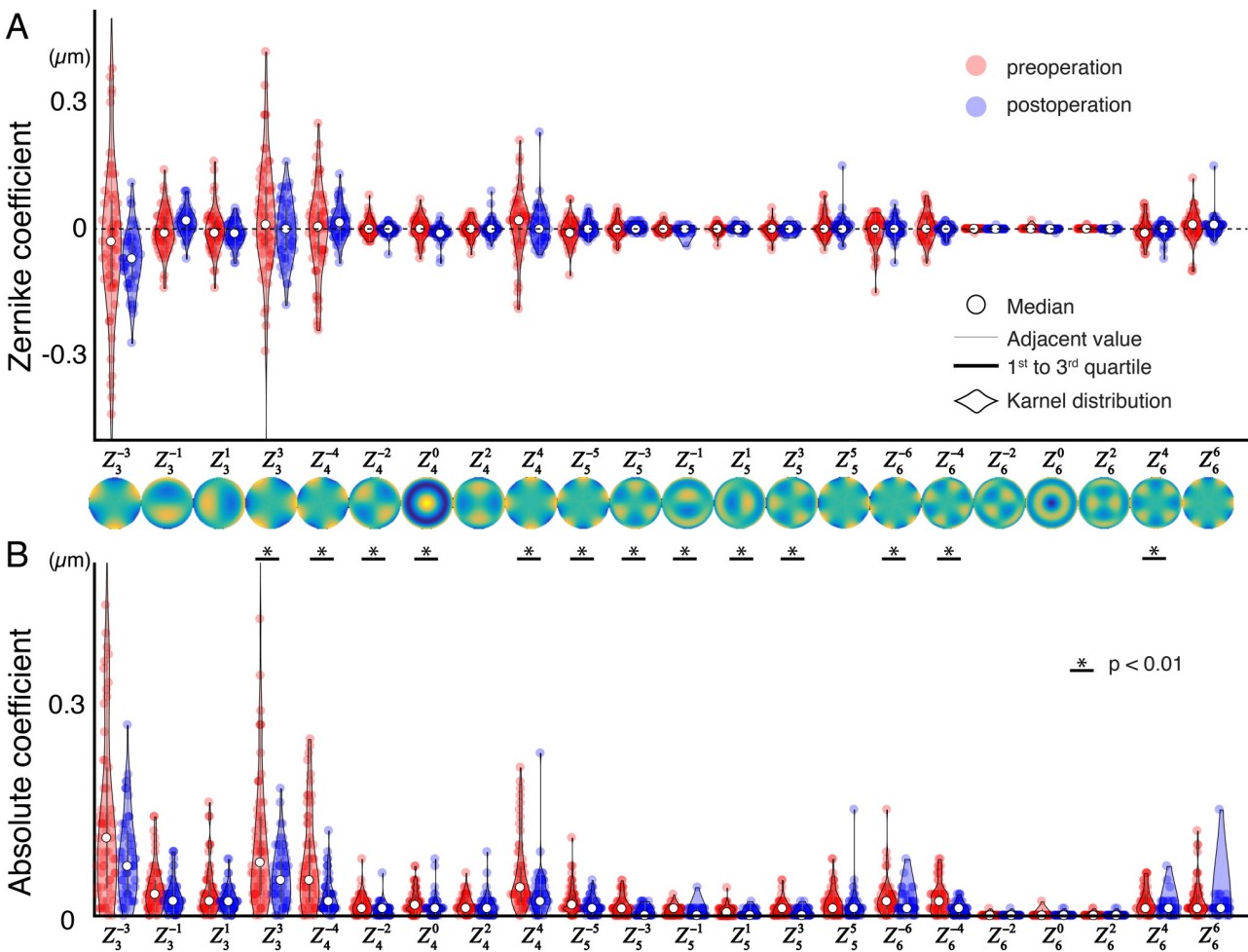

**Fig 2. Higher-order aberrations of the entire eye.** Data of 62 eyes in 62 patients are shown using violin plots. Wavefront aberrations of the entire eye measured by a retinoscopic aberrometry are decomposed by Zernike polynomials for a 3-mm pupil diameter. The vertical axis presents the values of Zernike coefficients (μm). The horizontal axis represents higher-order aberrations from the third (Z3) to sixth (Z6) orders of Zernike polynomials. A) Signed Zernike coefficients. The distributions of preoperative values of Zernike coefficients (red circles) clearly decreased in those of postoperative components (blue circles). B) Absolute Zernike coefficients. Preoperative values of Zernike absolute coefficients significantly decreased in more than half of postoperative components.

measurements, 32 cases remained as PP concerned cases, defined as patients either with pre- or postoperative photophobia scores of none zero. We calculated Spearman's rank correlation between photophobia score and Zernike coefficients of each Zernike term. There were significant associations between subjective photophobia scores and postoperative signed coefficients of $Z_4^2$, $Z_5^{-1}$ and $Z_6^4$ in all data ($r_s = 0.40$, -0.38 and -0.33) whereas there were no significant associations in the data set of all absolute cases ($p < 0.01$). However, moderate positive associations existed between subjective photophobia scores and postoperative absolute coefficients of $Z_3^{-1}$ ($r_s = 0.45$, $p = 0.01$) and $Z_4^4$ ($r_s = 0.42$, $p = 0.017$) in PP concerned patients (Table 2).

For RMS of wavefront errors in PP concerned patients, all eight combinations of photophobia scores and RMS data showed no significant correlation preoperatively (Table 3). Conversely, there were significant, moderate correlations between the subjective photophobia scores and THOA group ($r_s = 0.52$, $p = 0.002$), total coma group ($r_s = 0.52$, $p = 0.002$), total trefoil group ($r_s = 0.47$, $p = 0.006$), and third-order group ($r_s = 0.53$, $p = 0.002$) in postoperative PP concerned data (Fig 3). In the data set of all cases, which consisted of PP concerned and unconcerned patients, there was no significant correlation, but positive weak correlation existed only in RMS of total trefoil group ($r_s = 0.28$, $p = 0.027$) and third-order group ($r_s = 0.27$, $p = 0.03$). Because RMS of each combination in PP unconcerned patients were

**Table 2. Statistical correlation between photophobia scores and signed and absolute coefficients of each Zernike term.**

| Zernike term | Signed | | | | Absolute | | | |
|---|---|---|---|---|---|---|---|---|
| | Preoperative | | Post-operative | | Preoperative | | Post-operative | |
| | PP concerned | All data | PP concerned | All data | PP concerned | All data | PP concerned | All data |
| $Z_3^{-3}$ | 0.04 | 0.07 | −0.27 | −0.09 | −0.09 | 0.21 | 0.30 | 0.17 |
| $Z_3^{-1}$ | −0.15 | −0.06 | 0.34 | 0.23 | 0.05 | 0.00 | 0.45 | 0.12 |
| $Z_3^{1}$ | 0.06 | 0.08 | −0.16 | −0.11 | −0.04 | −0.25 | 0.34 | 0.02 |
| $Z_3^{3}$ | −0.12 | −0.20 | 0.10 | 0.03 | 0.10 | −0.03 | 0.31 | 0.31 |
| $Z_4^{-4}$ | −0.10 | 0.01 | 0.11 | 0.11 | 0.08 | −0.17 | 0.19 | 0.04 |
| $Z_4^{-2}$ | 0.08 | −0.07 | 0.09 | 0.10 | 0.07 | −0.06 | −0.12 | −0.01 |
| $Z_4^{0}$ | −0.04 | 0.01 | −0.14 | 0.00 | 0.12 | 0.06 | 0.06 | 0.11 |
| $Z_4^{2}$ | −0.14 | −0.22 | 0.01 | 0.40* | −0.11 | −0.16 | 0.32 | −0.05 |
| $Z_4^{4}$ | 0.36 | 0.14 | 0.36 | 0.29 | 0.16 | 0.06 | 0.42 | 0.15 |
| $Z_5^{-5}$ | −0.35 | −0.18 | −0.22 | −0.17 | 0.26 | −0.07 | −0.21 | −0.18 |
| $Z_5^{-3}$ | −0.04 | −0.12 | 0.26 | 0.18 | 0.07 | 0.26 | 0.11 | 0.14 |
| $Z_5^{-1}$ | 0.25 | 0.26 | −0.37 | −0.38* | −0.14 | 0.08 | 0.18 | 0.05 |
| $Z_5^{1}$ | −0.11 | −0.20 | −0.05 | −0.23 | −0.03 | −0.14 | 0.32 | 0.09 |
| $Z_5^{3}$ | 0.23 | 0.20 | 0.06 | 0.19 | 0.03 | −0.07 | 0.14 | 0.02 |
| $Z_5^{5}$ | −0.01 | −0.12 | −0.06 | 0.03 | 0.15 | 0.08 | 0.22 | −0.09 |
| $Z_6^{-6}$ | 0.02 | −0.06 | 0.10 | −0.07 | 0.25 | 0.18 | 0.09 | 0.22 |
| $Z_6^{-4}$ | 0.13 | −0.01 | −0.08 | −0.12 | 0.10 | −0.20 | 0.28 | 0.11 |
| $Z_6^{-2}$ | N/A | 0.12 | N/A | −0.12 | N/A | −0.12 | N/A | −0.12 |
| $Z_6^{0}$ | N/A | −0.12 | −0.12 | −0.05 | N/A | −0.12 | 0.90 | 0.20 |
| $Z_6^{2}$ | N/A | −0.17 | −0.02 | 0.16 | N/A | −0.17 | −0.02 | −0.01 |
| $Z_6^{4}$ | −0.35 | −0.08 | −0.31 | −0.33* | 0.23 | 0.04 | 0.26 | 0.12 |
| $Z_6^{6}$ | 0.13 | 0.01 | −0.12 | −0.02 | 0.31 | 0.02 | −0.13 | −0.02 |

N/A: not available,

* indicates statistically significant ($p < 0.01$).

Each value means $r_s$ (Spearman's rank sum correlation) between subjective photophobia scores and each signed and absolute Zernike coefficient (leftmost column) in each group (second row).

**Table 3. Statistical correlations between subjective photophobia scores and RMS of wavefront and other clinical parameters.**

| | Spearman rank sum correlation | | | |
|---|---|---|---|---|
| | Preoperative | | Post-operative | |
| | PP concerned | All data | PP concerned | All data |
| THOAs | 0.05 | −0.02 | 0.52* | 0.22 |
| Total coma | −0.01 | −0.13 | 0.52* | 0.11 |
| Total trefoil | 0.04 | 0.11 | 0.47* | 0.28 |
| TSA | 0.12 | 0.06 | 0.06 | 0.11 |
| 3rd | 0.03 | 0.07 | 0.53* | 0.27 |
| 4th | 0.00 | −0.16 | 0.21 | 0.03 |
| 5th | 0.28 | −0.04 | 0.05 | −0.07 |
| 6th | 0.24 | 0.01 | −0.03 | 0.17 |
| BCVA | 0.17 | −0.11 | 0.24 | −0.03 |
| Pupil size | −0.17 | −0.06 | −0.26 | −0.12 |
| M. Sphere | −0.04 | 0.11 | 0.17 | 0.17 |
| M. Cylinder | 0.36 | 0.14 | 0.11 | 0.18 |
| O. Sphere | 0.00 | 0.14 | 0.16 | 0.07 |
| O. Cylinder | 0.15 | 0.25 | 0.21 | 0.14 |
| Age | 0.07 | −0.10 | 0.32 | −0.01 |

THOA: total higher-order aberration, TSA: total spherical aberration, BCVA: best corrected visual acuity, M. Sphere: manifest spherical error, M. Cylinder: manifest cylindrical error, O. Sphere: objective spherical error, O. Cylinder: objective spherical error. Each value represents the $r_s$ value between subjective photophobia scores and each component of RMS or clinical parameter (leftmost column) in each group (second row), which was calculated by Spearman's rank sum correlation.
* means statistically significant ($p < 0.01$).

distributed widely, the correlation naturally became weaker in the data set of all patients. Additionally, we analyzed the association between subjective photophobia scores and other parameters obtained with normal ophthalmic measurements such as age, BCVA, spherical and cylindrical refractive errors, and pupil size (Table 3). There were no significant associations.

## Association between photophobia scores and HOAs or gender

The relationship between subjective photophobia score and RMS of HOAs (total coma and trefoil), age or gender was analyzed in a linear mixed-effect model. These results are shown in Table 4. According to this model, subjective postoperative photophobia score in PP concerned patients was positively associated with RMS of a total coma of 34.27 μm (95% CI, 10.42–58.13 μm; $p < 0.01$) and RMS of total trefoil of 13.4 μm (95% CI, 1.52–25.32 μm; $p < 0.05$). Subjective photophobia scores in females were higher than in males in three data sets; postoperative PP concerned patient, all postoperative patients and all pre- and postoperative all patients. In contrast, preoperative photophobia scores in all patients and PP concerned patients were not associated with all of 4 variables.

## Discussion

### Summary of results

In this study, we analyzed the changes in subjective photophobia scores before and after cataract surgery, and its association with wavefront aberrations. Our analysis showed that approximately half of the patients who underwent cataract surgery (35 of 71 patients) did not experience photophobia at all during the pre- and postoperative periods. Moreover, based on

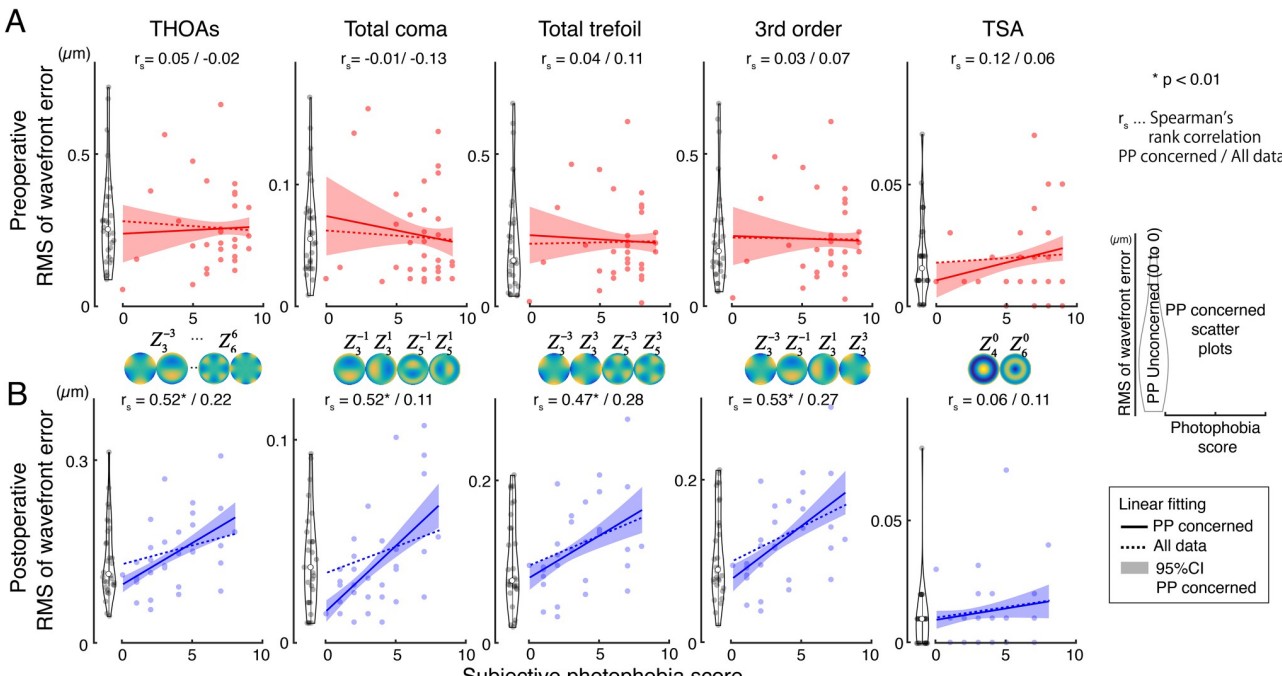

**Fig 3. Correlation between RMS of wavefront errors and photophobia score.** (A) Preoperative and (B) postoperative data are shown. The vertical axis represents positive root-mean-square (RMS) values of wavefront errors. The horizontal axis represents the subjective photophobia score. The distribution of RMS values in 30 cases (PP unconcerned) with a photophobia score of 0 both pre and postoperatively are shown using a violin plot on the left side of each graph. RMS data of other 32 cases (PP concerned) are plotted using red and blue circles. The solid line indicates linear fitting with PP concerned patients and dashed line indicates linear fitting with all of 62 eyes. The shaded area represents the 95% confidence interval of the linear fitting for PP concerned patients. Spearman's rank correlation ($r_s$) in the PP concerned cases and all cases is shown on the left and right side, respectively. (A) All five combinations of photophobia score and RMS data show no significant correlation. (B) Four components (THOAs: total higher-order aberrations, total coma, total trefoil, and third-order groups) show significant positive moderate correlations with photophobia score. TSA: total spherical aberrations.

our analysis, three-quarters of patients (27 of 36 patients) with cataract showed improvement in subjective photophobia after surgery—for those aware of the photophobia. HOAs, especially the third-order aberration including coma and trefoil aberrations, exhibited significant positive associations with subjective photophobia scores postoperatively, while components of wavefront aberrations showed no significant correlations with subjective photophobia scores preoperatively.

## Cataract surgery and subjective photophobia score

The definition of photophobia depends on the literature cited [41]. Some researchers defined it simply as visual discomfort due to bright light [20, 22], which could be considered photophobia in the broadest sense. In this definition, discomfort glare and photophobia are similar, as we described here. Another study defined photophobia as an abnormal response to normal illumination [21] and exposure of the eye to light definitely induces or exacerbates pain [23], which could be considered a more specific definition of photophobia. Photophobia in the more specific sense differs from discomfort glare in the point of the severity and threshold of occurrence. In this study, we simply defined photophobia as light-induced visual discomfort because there is no established definition for either normal illumination or light adaptation for healthy subjects and patients' threshold for discomfort have huge variability [18, 19].

**Table 4. Association between photophobia score and HOAs, age or gender with linear mixed effect model.**

| Photophobia score | | Variables | Co-efficient ± SE (95% CI) | p-Value |
|---|---|---|---|---|
| PP concerned patients (32 cases) | Preoperative | T. Coma | -10.40±12.20 (-35.42–14.62) | 0.40 |
| | | T. Trefoil | 1.83±3.46 (-5.27–8.93) | 0.60 |
| | | Age | 0.021±0.045 (-0.072–0.11) | 0.65 |
| | | Gender (F) | -0.042±0.85 (-1.78–1.70) | 0.96 |
| | Postoperative | T. Coma | 34.27±11.63 (10.42–58.13) | 0.0065** |
| | | T. Trefoil | 13.42±5.80 (1.52–25.32) | 0.029* |
| | | Age | 0.027±0.033 (-0.040–0.094) | 0.42 |
| | | Gender (F) | 1.74±0.57 (0.57–2.90) | 0.005** |
| | Both | T. Coma | 22.24±9.52 (3.20–41.29) | 0.61 |
| | | T. Trefoil | 6.30±3.46 (-0.63–13.22) | 0.023* |
| | | Age | 0.037±0.027 (-0.018–0.091) | 0.074 |
| | | Gender (F) | 1.32±0.48 (0.36–2.27) | 0.18 |
| All patients (62 cases) | Preoperative | T. Coma | -4.46±13.58 (-31.66–22.74) | 0.74 |
| | | T. Trefoil | 1.13±3.22 (-5.32–7.58) | 0.73 |
| | | Age | -0.034±0.054 (-0.14–0.074) | 0.53 |
| | | Gender (F) | 0.80±0.98 (-1.16–2.77) | 0.42 |
| | Postoperative | T. Coma | 18.77±12.27 (-5.82–43.35) | 0.13 |
| | | T. Trefoil | 15.88±5.65 (4.57–27.19) | 0.007** |
| | | Age | -0.032±0.034 (-0.010–0.036) | 0.35 |
| | | Gender (F) | 1.46±0.56 (0.33–2.59) | 0.012* |
| | Both | T. Coma | 16.44±7.49 (1.61–31.28) | 0.030* |
| | | T. Trefoil | 4.50±2.01 (0.51–8.49) | 0.028* |
| | | Age | -0.002±0.033 (-0.067–0.063) | 0.95 |
| | | Gender (F) | 1.18±0.57 (0.049–2.32) | 0.041* |

T. Coma: RMS of total coma aberration, T. Trefoil: RMS of total trefoil aberration, Gender (F): Female. Both: both preoperative and postoperative photophobia scores.

* and ** means statistically significant (p < 0.05 and p < 0.01, respectively).

Glare is one of the major symptoms associated with cataract. Cataract surgery practically reduces disability glare [42, 43] because the abnormal light stimuli to the retina due to the lens imperfection, represented by forward scattered light—source of the disability glare [14]—is removed. However, with limited reports on the association between cataract and photophobia, removing the cloudy lens and implantation of a clear IOL affect the spatial pattern and spectral distribution of the retinal inputs. Bargary et al. reported that discomfort glare is more closely associated with the spatial properties of the glare source [44], indicating that scattering light could evoke not only disability but also discomfort glare; thus, cataract surgery could reduce photophobia. Meanwhile, short wavelength light evokes photophobia more effectively than longer wavelength light [20], which suggests that cataract surgery could easily evoke photophobia because both clear and tinted IOLs have higher transmittance at the short wavelength than that of the human crystalline lens in middle aged and older individuals [10, 45, 46].

Our study evaluated photophobia associated with cataract using a subjective photophobia score similar to the pain scaling score [34], which is a classical and reliable method. To the best of our knowledge, this is the first study that evaluated light-induced visual discomfort before and after cataract surgery using a score scale. Our data showed improvement of photophobia score after surgery in 27 (75.0%) of the 36 PP concerned patients (excluding a patient who scored 0 both before and after surgery), which suggests that cataract surgery improves photophobia by decreasing the spatial property of the retinal inputs beyond the increasement of

light inputs, especially for short wavelength light. Half of our patients with cataract were not in any way bothered by photophobia either before or after cataract surgery, which is consistent with a previous study that showed the major reason for receiving cataract surgery was the decrease in visual acuity with episodes such as difficulty of driving at night, reading fine print, and handicraft (as in tailoring) [47].

## HOAs and subjective photophobia score

With PSF, the light signal incident at the cornea (input) is optically transformed into the image formed at the retina (output) [48]. The PSF of the human eye consists of two extremely different domains [9]. One is a straylight, which is estimated psychophysically and has the potential to cause both disability [14] and discomfort glare [19]. The other is a PSF core, which is physically measurable using the double-pass method [26] and calculated from wavefront aberrations [49]. HOAs are defined as the wavefront aberration excluding refractive errors of spheres and cylinders [27]. Visual improvement by HOA correction had been predicted theoretically [27] and HOAs practically affect contrast sensitivity [29, 30], high-contrast visual acuity [31], low-contrast visual acuity [28], depth of focus [50], and visual symptoms such as double vision and starburst [51, 52]. In this study, we focused on HOAs and investigated their association with light-induced visual discomfort before and after cataract surgery. The preoperative HOAs in our study showed the cataract-specific pattern as each component distributed both positively and negatively evenly. After surgery, more than half of HOA components significantly reduced. These pre- and postoperative HOA findings in our study are consistent with those in previous studies [53, 54].

Why would increasing HOAs affect the degree of photophobia? Increments of HOAs change the retinal spatial property—particularly high contrast patterns such as around a light source. We calculated the PSFs from HOAs in the ideal observer (no HOAs) and representative patients (Fig 4). There was no obvious difference in PSF between the aberration-free ideal observer (the left column) and all participants at 550 nm. However, as the wavelength was shorter (500nm), the PSFs of two of the left columns (the ideal observer and the subject with small aberrations) and two of the right columns (those of the two participants with large higher-order aberrations and high subjective PP score) differ greatly. It is well known that spatial property largely affects lightness perception [55, 56] and makes an observer overestimate the brightness of an object [57]. For processing brightness and color information, the ventral occipital lobe plays an important role [58–60] and a case series reported that bilateral ventral occipital lesions caused a lack of photophobia [61].

Consequently, changing retinal properties based on HOAs might cause photophobia based on hyperexcitability in such cortical areas [44]. Additionally, alteration of the yellowish crystalline lens to clear intraocular lens by cataract surgery increases shorter wavelength light which facilitates photophobia. That PSFs at shorter wavelengths were more affected with HOAs, as shown here, may also increase subjective photophobia scores. Notably, our results show that no component of preoperative HOAs was significantly associated with photophobia score and could not be explained by the previous interpretation using retinal inputs and HOAs, although we analyzed with not only single linear regression but also multivariate analysis. We considered the results implicitly supporting the influence of another factor not measured—forward scattering. Thus, we interpreted that photophobia induced by forward scattered light preoperatively masked the photophobia induced by HOAs, which may be relatively small, in cataracts. Our data suggested that photophobia due to HOA may partly explain photophobia in patients with IOL. The development of IOLs with less HOA abnormalities will lead to satisfaction with cataract surgery by solving the complication of postoperative photophobia.

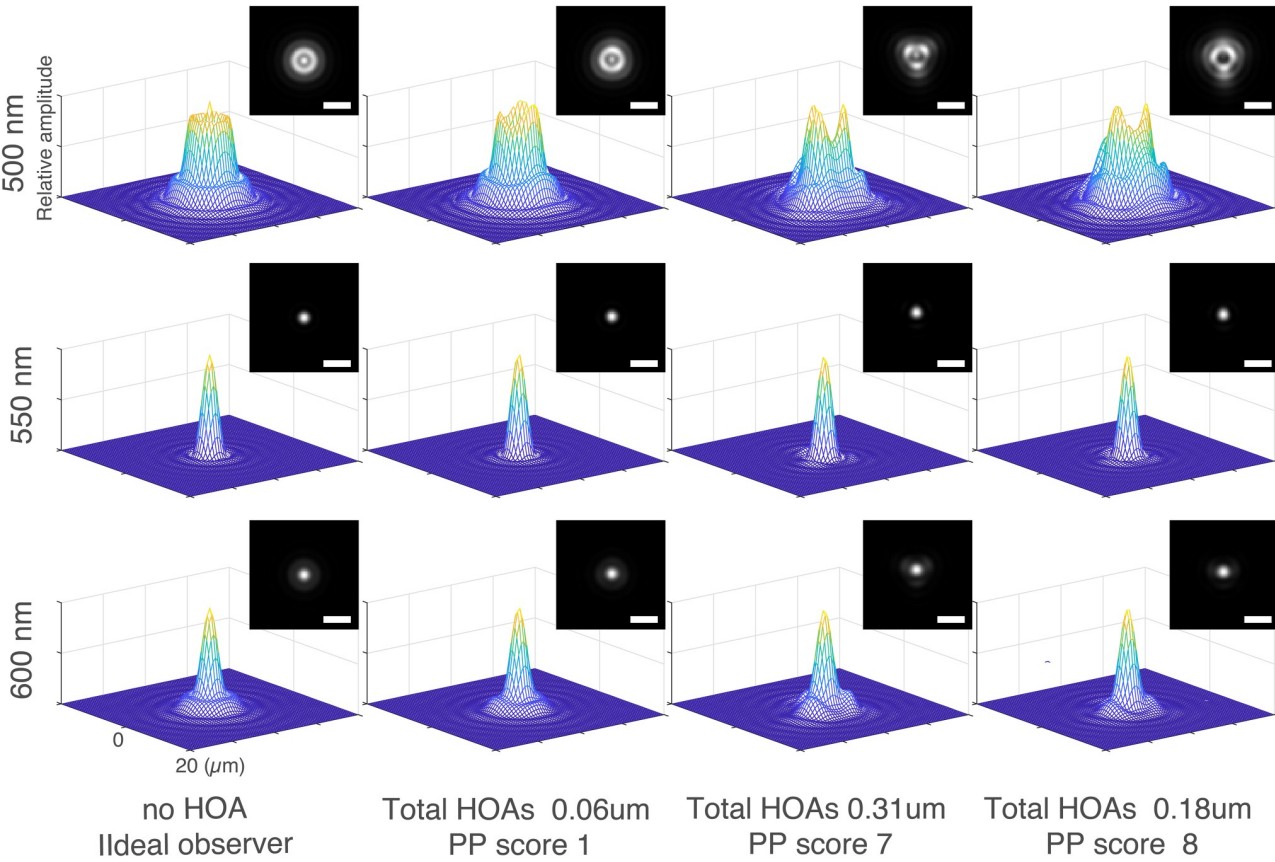

**Fig 4. Point spread function from HOAs of ideal observer and representative patients.** The PSFs of the ideal observer (the left column, no HOAs) and representative patients in this study are estimated with 3mm diameter pupil from higher-order aberrations at 500 nm (top row), 550 nm (middle row), and 600 nm (bottom row), respectively. All PSFs are normalized to the peak of each amplitude. The upper right inlet shows 2D image of each PSF. The scale bar is 10 μm.

## Non-visual factors and subjective photophobia score

We have discussed photophobia induced by visual input. However, other neural inputs, namely trigeminal inputs, have long been considered to contributed to photophobia [23]. Anterior eye diseases such as keratitis and iritis can cause photophobia [22]. In addition, there are several reports of photophobia enhanced by migraine [62–64] associated with trigeminal nerve input. In rodents, cells that receive trigeminal and visual inputs were found in the posterior thalamus [65]. Furthermore, the association between inputs to the ipRGC with a strong response to short-wavelength light and photophobia during migraine has been reported in humans [34]. Thus, pain and photophobia are considered strongly related sensations. Gender is an important factor in pain sensitivity differences [66]. Females are 2–3 times more likely than males to have migraines due to hormonal differences [67] while no gender differences exists in photophobia in visually normal people [36], and males have higher brightness perception than discomfort [35]. In our results, females had significantly higher subjective photophobia scores than males, suggesting the influence of postoperative subthreshold trigeminal inputs that a patient never perceive because females demonstrate heightened central sensitization [68]. Visual photosensitivity threshold was reported to increases with age in visually normal subjects [36]. Although the effect of age was not observed in photophobia scores in our data,

this might be because patients underwent surgery for visual impairment due to cataracts; thus were older and had a relatively narrower range of age.

## Limitations

There are limitations to this study. Selection bias cannot be excluded because this study targeted patients in the narrow geographic region from a single center. Additionally, the main analysis in our study targeted the relationship between subjective photophobia scores and HOAs. However, other important factors associated with photophobia have been reported such as forward or backward scattered light. Furthermore, the subjective photophobic score was determined using the recall method, which seemed to not be efficient for approximately half of the patients who scored zero both pre- and postoperatively. Although an objective measurement for the threshold and degree of photophobia has not been established, the threshold of photophobia could be physically defined in a particular condition. Additionally, the follow-up period was relatively short because most visual disturbance resolves in the first year postoperatively. However, it is difficult to follow-up the patient who does not worry since their symptoms have resolved. Further large-scale, longitudinal follow-up studies covering the data of multi-modalities, including threshold measurements for photophobia and information about the degree of ocular pain and headaches, are required to further explore these relationships.

## Conclusions

Our results showed that approximately half of the patients who require cataract surgery did not experience any subjective photophobia both before and after surgery. Contrastingly, among the patients who reported photophobia before surgery, three-quarters reported improvement in subjective photophobia after surgery. Lastly, our data indicated that HOAs induced by the cornea and IOLs may explain part of the subjective postoperative photophobia.

## Supporting information

**S1 Text. Photophobia scaling score.**
(DOCX)

## Acknowledgments

The authors would like to thank Hiroshi Tsuneoka, Satoshi Nakadomari and Ryo Terauchi for supporting this study and the three anonymous reviewers for their insightful suggestions and careful reading of the manuscript.

## Author Contributions

**Conceptualization:** Hiroshi Horiguchi, Satoshi Katagiri, Takuya Shiba.

**Data curation:** Naoko Ishiguro, Hiroshi Horiguchi.

**Formal analysis:** Naoko Ishiguro, Hiroshi Horiguchi.

**Funding acquisition:** Hiroshi Horiguchi.

**Investigation:** Naoko Ishiguro, Hiroshi Horiguchi, Satoshi Katagiri.

**Methodology:** Hiroshi Horiguchi.

**Project administration:** Satoshi Katagiri, Takuya Shiba, Tadashi Nakano.

**Resources:** Tadashi Nakano.

**Supervision:** Takuya Shiba, Tadashi Nakano.

**Validation:** Takuya Shiba, Tadashi Nakano.

**Visualization:** Naoko Ishiguro, Hiroshi Horiguchi.

**Writing – original draft:** Naoko Ishiguro, Hiroshi Horiguchi, Satoshi Katagiri.

**Writing – review & editing:** Naoko Ishiguro, Hiroshi Horiguchi, Satoshi Katagiri, Takuya Shiba, Tadashi Nakano.

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
