## [Decision Letter · Decision Letter 0]

15 Jun 2022

PONE-D-22-10607Correlation between higher-order aberration and photophobia after cataract surgeryPLOS ONE

Dear Dr. Horiguchi,

Thank you for submitting your manuscript to PLOS ONE. After careful consideration, we feel that it has merit but does not fully meet PLOS ONE’s publication criteria as it currently stands. Therefore, we invite you to submit a revised version of the manuscript that addresses the points raised during the review process.

Please implement all changes requested by the reviewers.

We look forward to receiving your revised manuscript.

Kind regards,

Manuel Spitschan

Academic Editor

PLOS ONE

Journal Requirements:

"Our study was supported by the grant of Japan Society for the Promotion of Science (JSPS) KAKENHI (JP18K16939 and 21K09729 to H.H.) and the Charitable Trust Fund for Ophthalmic Research in Commemoration of Santen Pharmaceutical’s Founder (H.H.)"

"The authors received no specific funding for this work."

Reviewers' comments:

Reviewer's Responses to Questions

**Comments to the Author**

1. Is the manuscript technically sound, and do the data support the conclusions?

Reviewer #1: Yes

Reviewer #2: Yes

2. Has the statistical analysis been performed appropriately and rigorously? 

Reviewer #1: Yes

Reviewer #2: Yes

3. Have the authors made all data underlying the findings in their manuscript fully available?

Reviewer #1: No

Reviewer #2: Yes

4. Is the manuscript presented in an intelligible fashion and written in standard English?

Reviewer #1: Yes

Reviewer #2: Yes

5. Review Comments to the Author

Reviewer #1: Review of PONE-D-22-10607

In this well-written paper the authors describe a relationship between photophobia and higher order lens aberrations measured before and after cataract surgery. They show an anticipated reduction in glare following surgery, and a correlation between glare and higher order lens aberrations.

The authors need to say a little more about how they measured photophobia. “We scored photophobia from 0 (patient does not experience photophobia at all) to 10 (patient experiences severe photophobia to open eyes) … We asked patients to score the degree of photophobia on a sunny day…” There appears to have been little control over the viewing conditions, indeed patients appear to be scoring the photophobia from memory. Please specify exactly how photophobia was measured i.e. the wording of the questions used and the setting in which they were asked.

The authors acknowledge that large individual differences in self-reported glare occur despite identical “visual inputs”. Some individuals are more susceptible to uncomfortable sensations than others, and this is often related to headache susceptibility. Do the authors have information concerning headaches? Because the susceptibility to uncomfortable sensation varies with individual, the authors are correct to analyse the worst eye, but they refer to “35 of 41 eyes” as if the eyes are independent of the individual, which they are not.

The distinction between glare and photophobia needs to be elaborated a little. In particular it would be useful to know whether patients experienced pain. Photophobia often implies discomfort or pain, whereas glare usually implies impaired vision. The distinction is often made in terms of “discomfort glare” and “disability glare”. Were the patients reporting pain (photophobia) or simply glare? I suspect when the scores were low they were reporting disability glare, and when high both disability and discomfort glare. The authors have divided scores into zero and above zero. Is this an appropriate division? Were the high scores associated with headaches?

There are a few minor infelicities of English usage as follows:

“impinged spatial properties and wavelength distribution on retinal images.”

“decreased in postoperatively”

“alternation in photophobia” instead of alteration in photophobia

Reviewer #2: The manuscript describes a technically reliable scientific study with data supporting the conclusion. The study was carried out strictly with proper control, replication and sample size. This study draws appropriate conclusions based on the data provided.

6. PLOS authors have the option to publish the peer review history of their article (what does this mean?). If published, this will include your full peer review and any attached files.

Reviewer #1: No

Reviewer #2: No

---

## [Author Response · Author response to Decision Letter 0]

24 Jul 2022

We uploaded a word document; Response to Reviewer. Please read the document.

---

## [Editor Report · Decision Letter 1]

2 Sep 2022

Correlation between higher-order aberration and photophobia after cataract surgery

PONE-D-22-10607R1

Dear Dr. Horiguchi,

We’re pleased to inform you that your manuscript has been judged scientifically suitable for publication and will be formally accepted for publication once it meets all outstanding technical requirements.

Kind regards,

Manuel Spitschan

Academic Editor

PLOS ONE
---

## [Editor Report · Acceptance letter]

6 Sep 2022

PONE-D-22-10607R1 

Correlation between higher-order aberration and photophobia after cataract surgery 

Dear Dr. Horiguchi:

I'm pleased to inform you that your manuscript has been deemed suitable for publication in PLOS ONE. Congratulations! Your manuscript is now with our production department. 

Kind regards, 

on behalf of

Dr. Manuel Spitschan 

Academic Editor

PLOS ONE